# Induction of apoptosis by double-stranded RNA was present in the last common ancestor of cnidarian and bilaterian animals

Itamar Kozlovski[1]☯*, Adrian Jaimes-Becerra[1]☯, Ton Sharoni[1], Magda Lewandowska[1], Ola Karmi[2], Yehu Moran[1]*

1 Department of Ecology, Evolution and Behavior, Alexander Silberman Institute of Life Sciences, Faculty of Science, The Hebrew University of Jerusalem, Jerusalem, Israel, 2 Research Infrastructure Facility, Alexander Silberman Institute of Life Sciences, Faculty of Science, The Hebrew University of Jerusalem, Jerusalem, Israel

☯ These authors contributed equally to this work.
* itamar.kozlovski@mail.huji.ac.il (IK); yehu.moran@mail.huji.ac.il (YM)

**Data Availability Statement:** All data used in this work is either included in the main text, figures, supplementary files and available at Genbank,

## Abstract

Apoptosis, a major form of programmed cell death, is an essential component of host defense against invading intracellular pathogens. Viruses encode inhibitors of apoptosis to evade host responses during infection, and to support their own replication and survival. Therefore, hosts and their viruses are entangled in a constant evolutionary arms race to control apoptosis. Until now, apoptosis in the context of the antiviral immune system has been almost exclusively studied in vertebrates. This limited phyletic sampling makes it impossible to determine whether a similar mechanism existed in the last common ancestor of animals. Here, we established assays to probe apoptosis in the sea anemone *Nematostella vectensis*, a model species of Cnidaria, a phylum that diverged approximately 600 million years ago from the rest of animals. We show that polyinosinic:polycytidylic acid (poly I:C), a synthetic long double-stranded RNA mimicking viral RNA and a primary ligand for the vertebrate RLR melanoma differentiation-associated protein 5 (MDA5), is sufficient to induce apoptosis in *N. vectensis*. Furthermore, at the transcriptomic level, apoptosis related genes are significantly enriched upon poly(I:C) exposure in *N. vectensis* as well as bilaterian invertebrates. Our phylogenetic analysis of caspase family genes in *N. vectensis* reveals conservation of all four caspase genes involved in apoptosis in mammals and revealed a cnidarian-specific caspase gene which was strongly upregulated. Altogether, our findings suggest that apoptosis in response to a viral challenge is a functionally conserved mechanism that can be traced back to the last common ancestor of Bilateria and Cnidaria.

## Author summary

Apoptosis is a programmed cell death mechanism used by vertebrates to efficiently block viral infection. The presence of long double-stranded RNA (dsRNA) in the cytosol is a key feature of DNA and RNA virus replication and is absent from uninfected mammalian

Uniprot and genome repositories. Accession numbers are included in S2 Table. All raw data required to replicate the results of this study is available in S4–S7 Tables.

**Funding:** This work was supported by a European Research Council Consolidator grant (863809) to YM. Funder URL: https://erc.europa.eu/homepage. The funder did not play any role in study design, data collection and analysis, decision to publish, or preparation of the manuscript.

**Competing interests:** The authors have declared that no competing interests exist.

host cells. Therefore, the ability to sense and respond to viral dsRNAs is crucial for organismal survival. Indeed, numerous studies in mammals have shown that dsRNA has the capacity to trigger a robust apoptosis as antiviral response. However, such studies were limited to vertebrates, and it remained largely unclear how such systems have evolved. Here, we used the sea anemone *Nematostella vectensis*, member of phylum Cnidaria (jellyfish, corals, hydroids and sea anemones), to address this topic. We demonstrated that indeed dsRNA is sufficient to induce apoptosis in *Nematostella* and uncovered a conserved network of genes involved in this process. Further, by comparing the results of gene expression analyses in sea anemones and other diverse animal groups such as oysters and lancelets, we show that apoptosis is prevalent in many animal groups and was already part of the response to dsRNA in the last common ancestor of Cnidaria and most other animals that lived 600 million years ago.

## Introduction

Apoptosis, a major form of programmed cell death, is one of the most important innovations of multicellular eukaryotes, and its components, at least in some form, are found in representatives of all multicellular eukaryotes [1]. The field of apoptosis research has thrived since the discovery of its molecular mechanism in the nematode *Caenorhabditis elegans* [2,3]. Since then, apoptosis has been extensively studied in mammals, insects, and nematodes, and has also been described in some of the most basally-branching animal lineages, such as sponges and cnidarians [4–11]. Furthermore, processes similar to apoptosis have also been observed in plants [12], fungi [13–15], and even in some unicellular eukaryotes [16–19]. In bacteria, other forms of regulated cell death such as pyroptosis have been described [20] whereas diverse homologs of key components of the apoptotic machinery have been found in bacterial genomes [1,21]. Interestingly, recent studies have demonstrated that bacteria employ regulated forms of cell death in response to phage infection, thereby protecting the nearby population from the spread of infection [22]. These studies suggest that some of the prokaryotic defense systems involving regulated cell death are evolutionarily ancestral of key components of eukaryotic immune systems and that regulated cell death is an efficient mechanism for blocking viral infection in all domains of life.

In their struggle for existence, viruses and hosts have evolved diverse mechanisms to control and manipulate the apoptotic pathway. Cell death is important for limiting pathogen replication in infected cells, while simultaneously promoting the inflammatory and innate immune responses that are crucial for host immunity [23]. Since viruses are obligatory intracellular parasites, they must be able to manipulate the apoptotic pathways in order to control the lifespan of their host and to replicate. Evidently, many viruses encode inhibitors of apoptosis to evade host responses during infection and to support their own replication and survival [24]. Dysregulated cell death as a result of defects in the apoptotic pathways is often a characteristic of inflammatory, autoimmune disorders, and cancer [25–27]. Therefore, as both viruses and cells struggle for control of cell death pathways, evolutionary arms race has shaped a complex host-pathogen interaction at this front [28].

Apoptosis has morphological characteristics that include cell shrinkage, nuclear fragmentation, chromatin condensation and membrane blebbing, all of which are the result of the proteolytic activity of the caspase proteases [29,30]. Upon activation, caspases induce apoptosis through the cleavage of several proteins, eventually leading to the phagocytic recognition and engulfment of the dying cell [31]. In vertebrate cells, apoptosis typically proceeds through one

of two signaling pathways termed the intrinsic and extrinsic pathways, both of which result in activation of the executioner caspases, Caspase-3 and Caspase-7 [31]. The intrinsic pathway is initiated by cell stress that is sensed by internal sensors such as p53, whereas the extrinsic pathway is initiated by an external stimulus such as death ligand/death receptor (for example, TNF and TNF receptor) interactions [24].

Double-stranded RNA (dsRNA) is associated with most viral infections, either directly by the viral genome itself (in the case of dsRNA viruses) or indirectly by generating intermediates in host cells during viral replication [32,33]. In mammals, a multitude of studies demonstrated that dsRNA is able to activate a strong antiviral response through the stimulation of type I IFN (IFN-I), and to inhibit the growth of mouse tumors [34,35]. In vertebrates, viral RNAs are recognized by intracellular pattern recognition receptors (PRRs) such as the RIG-I-like receptors (RLRs) retinoic acid-inducible gene I (RIG-I) and melanoma differentiation-associated protein 5 (MDA5) [36–38]. Upon recognition of their ligands, RIG-I and MDA5 interact with the adaptor protein mitochondrial anti-viral signaling protein (MAVS), which provides a scaffold to activate the NF-κB and IRF3 transcription factors [39]. Both transcription factors, in turn, regulate the expression of genes that can initiate the apoptotic pathway [28,40].

Until now, apoptosis in the context of the antiviral immune system has been almost exclusively studied in vertebrates. From this limited phyletic sampling, it is impossible to deduce what was the original mode of action of the antiviral response in the last common ancestor of vertebrates and whether apoptosis played a role in antiviral immunity in early-branching animal lineages. The phenomenon of bleaching in Cnidaria (sea anemones, corals, hydroids and jellyfish) has motivated several groups to address questions regarding apoptosis. In the stony coral *Stylophoea pistillata*, it was demonstrated that bleaching and death of the host animal in response to thermal stress involves a caspase-mediated apoptotic cascade induced by reactive oxygen species produced primarily by the algal symbionts [41]. A study of the stony corals *Pocillopora damicornis* and *Oculina patagonica* demonstrated that reduced pH conditions induce tissue-specific apoptosis that leads to the dissociation of polyps from coenosarcs [42]. Another study demonstrated that coral cells undergo apoptosis in response to human TNFα and that coral TNF kills human cells through direct interaction with the death receptor pathway [43]. Pyroptosis, a regulated form of inflammatory cell death that involves plasma membrane pore formation mediated by gasdermin family proteins [44], was recently reported in response to lipopolysaccharide (LPS) in *Hydra vulgaris* [45] and in response to the bacterial pathogen *Vibrio coralliilyticus* in the reef-building coral species *Pocillopora damicornis* [46]. However, the role of apoptosis or other forms of cell death as antiviral mechanisms in invertebrates remains mostly unexplored.

Over the past two decades the sea anemone *N. vectensis* has gained popularity as a model organism for studying molecular evolution, traditionally in the context of development and regeneration [47,48]. As a cnidarian, *N. vectensis* is particularly informative for comparative biology. This is because Cnidaria is a sister group to Bilateria (the group including the vast majority of extant animals) and these two groups diverged approximately 600 million years ago from their last common ancestor [47,49,50]. We have previously shown that *N. vectensis* possesses two RLR paralogs (named RLRa and RLRb) of the mammalian MDA5 that senses dsRNA. Further, we showed that RLRb binds to long dsRNA to initiate a functionally conserved innate immune response [51]. In the current study, we sought to determine whether and how dsRNA affects apoptosis in *N. vectensis*. We found that the dsRNA viral mimic poly (I:C) is sufficient to strongly induce apoptosis in *N. vectensis* and uncovered a gene network involved in this process in *N. vectensis* and various bilaterian groups, pointing to a functional conservation of dsRNA-induced apoptosis that existed before the cnidarian-bilaterian split.

## Results

### dsRNA increases the expression of genes associated with the apoptotic pathway

To further investigate the impact of dsRNA on apoptosis, we reanalyzed a gene expression dataset previously published by Lewandowska *et al.* (2021) [51], which examined the antiviral immune response of *N. vectensis* to a viral dsRNA mimic (poly(I:C)). Compared to the 0.9% NaCl control, dsRNA induced a gene expression profile strongly associated with the apoptotic process. This included increased expression of genes encoding initiator and executioner caspases (Caspase-8 and Caspase-3/6/7-like1, with Log$_2$ fold change (LFC) of 2.84 and 3.37, respectively), negative apoptotic regulators (Bcl-2-like3, MCL and IAP, with LFC of 1.39, 2.86 and 1.4, respectively), and positive apoptotic regulators (Bok, Apaf-1 and TRAF3-like1, with LFC of 3.63, 1.12 and 1.85) at 24 hours post injection (hpi) (**Fig 1A** and **S1 Table: Sheets 1 and 3**). However, this upregulation was temporary, as a decline in the number of apoptosis-related genes being upregulated was noted by 48 hpi (**Fig 1A**, **S1 Table: Sheet 4**). 34 upregulated genes were detected at 6 hpi and none of them were related to apoptosis (**S1 Table: Sheet 2**). Gene ontology (GO) term analysis revealed significant enrichment of apoptosis-related terms induced by dsRNA (**Fig 1B**), with the term 'Regulation of apoptotic process' being most significant at the 24 hpi time point. Furthermore, based on the identified upregulated genes associated with apoptosis in *N. vectensis* at 24 hpi, it seems that both the intrinsic and the extrinsic pathways of apoptosis were activated. Considering only the upregulated genes associated with apoptosis identified at 24 hpi, we propose and illustrate a hypothetical model for both the intrinsic and extrinsic pathways of apoptosis in *N. vectensis*. This model is based on homology and existing knowledge from the bilaterian system (**Fig 1C**). The intrinsic pathway (**Fig 1C**) may be initiated by various intracellular signals, such as DNA damage. These signals lead to the downregulation of pro-survival proteins like Bcl-2 and Bcl-10, thereby enabling pro-apoptotic molecules such as Bok (At 24 hpi, a homolog was found to be upregulated in *N. vectensis*, with a LFC value of 3.63), to provoke a mitochondrial permeability transition. This transition results in the release of cytochrome c and other pro-apoptotic molecules, counteracting the anti-apoptotic activity of IAPs [52]. Following this, cytochrome c associates with Apaf-1 to form the apoptosome (At 24 hpi, a homolog was found to be upregulated in *N.vectensis*, with a LFC value of 1.12*)*, which activates initiator caspases (At 24 hpi, one initiator caspase was found to be upregulated in *N. vectensis*. Caspase-8 with a LFC value of 2.8). Subsequently, initiator caspases trigger the executioner caspase cascade, culminating in cell death (At 24 hpi, two homologs of executioner caspases were found to be upregulated. Caspase 3/6/7-like1 and Caspase 3/6/7-like2, with LFC values of 3.37 and 1.15 respectively). In contrast to the intrinsic pathway, the extrinsic pathway (**Fig 1C**) is initiated by the binding of secreted ligands from the TNF family to their corresponding receptors within the TNFR family (At 24 hpi, homologs of TNF receptor-associated factor were found to be upregulated. TRAF3-like1 and TRAF3-like2, with LFC values of 1.85 and 2.62 respectively). The sole distinction between the two pathways is that the extrinsic pathway involves death receptors at the initiation of the apoptotic cascade [52]. Upon activation, receptor homotrimers or homomers recruit cytoplasmic adaptor molecules, which then instigate cell death through various pathways. Following the activation of classical cell death receptors (e.g., FAS, DR4/5, TNFR1), adaptor molecules directly bind and facilitate the cleavage of initiator caspases. These, in turn, activates the executioner caspases that orchestrate cell death [53]. Importantly, other types of programmed cell death, such as necroptosis and pyroptosis, have been reported in the context of the antiviral immune response in mammals [54–56]. Nonetheless, aside of caspase-8, we did not identify any human homologous components of necroptosis in the *N.vectensis* genome and transcriptome (**Fig 2**). We did find a homolog (NVE20414) of Gasdermin E, a precursor of a pore-forming protein which mediates

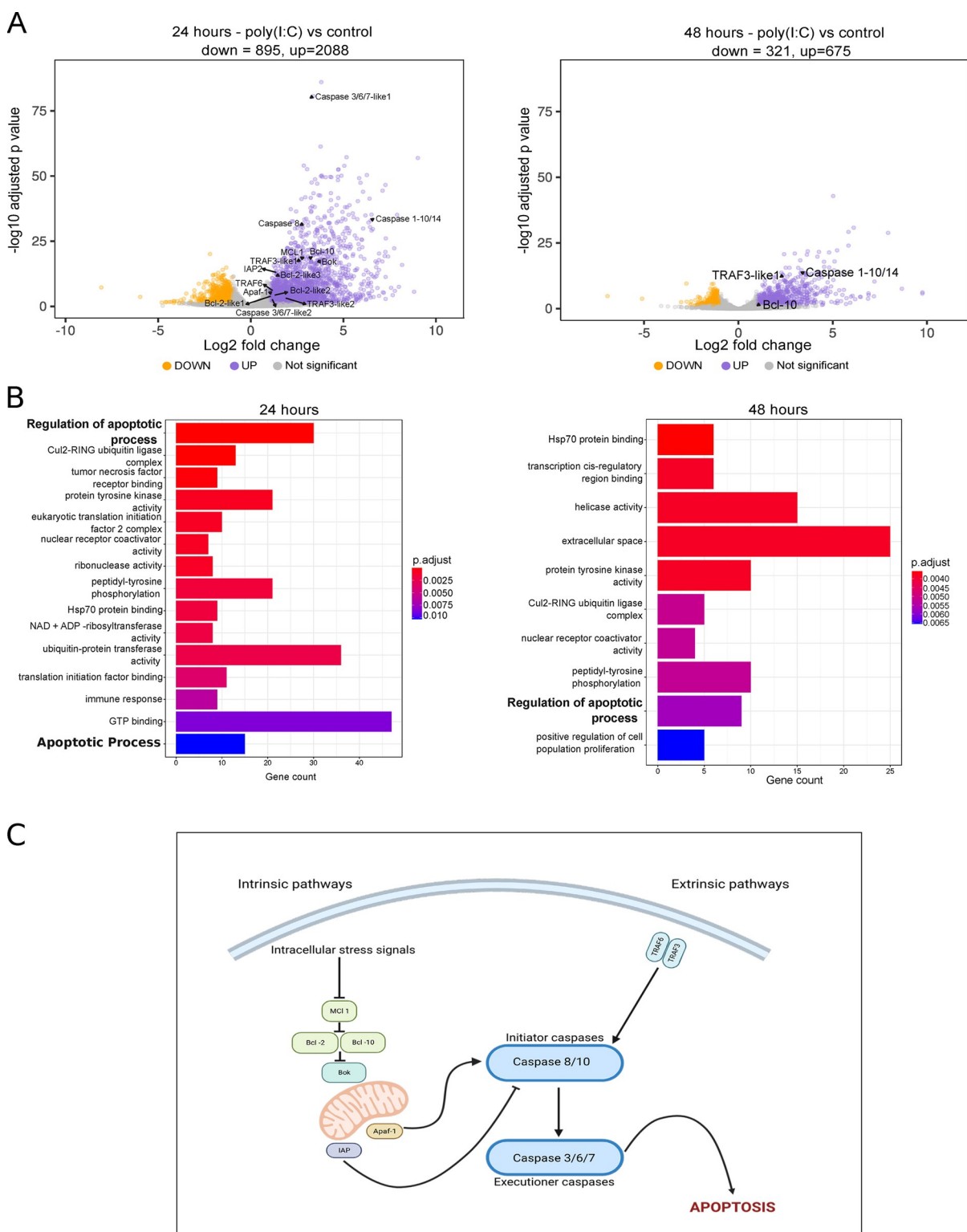

**Fig 1. Micronjection of the dsRNA viral mimic poly(I:C) induces expression of apoptosis pathway genes.** The dataset analyzed in this study was sourced from Lewandowska et al., 2021 [51]. (A) Volcano plots showing differentially expressed genes (DEGs) in *N. vectensis* after poly(I:C) administration at 24 hpi and 48 hpi. Significantly up- and downregulated genes (adjusted p < 0.05 and absolute log₂ fold change > 1) are highlighted in purple and orange, respectively. Among those, genes associated with the apoptosis pathway are highlighted, along with their names. (B) The bar plots illustrate enriched Gene Ontology (GO) terms, showcasing the Top 15 and Top 10 overrepresented

biological process GO terms among upregulated genes at the 24 hpi and 48 hpi time points, respectively. All terms displayed are statistically significant as per the hypergeometric test (with a Benjamini-Hochberg corrected p ≤ 0.05). (C) A simplified schematic illustrates the proposed intrinsic and extrinsic apoptotic signaling pathways in *N. vectensis* based on the upregulated components upon poly(I:C) administration at 24 hpi which possess homologs in their metazoan counterparts.

pyroptosis in mammals [57,58], to be upregulated in our transcriptomic data (LFC = 1.8, p = 0.0014), suggesting that pyroptosis might also be activated. However, we could not identify homologs for Caspase-1 and NLRP3 in the *N. vectensis* genome, which are crucial for inducing pyroptosis in mammals [59,60], in our genomic and transcriptomic analyses (**Fig 2**).

## dsRNA induces apoptosis in *N. vectensis* derived cells *ex-vivo*

We used Apotracker Green (Apo-15), an Annexin V based marker which is commonly used in mammalian cells to probe apoptosis by flow cytometry [61,62]. We first used either mitomycin-c (MMC) or cycloheximide (CHX) treatments, common inducers of apoptosis in human and mouse cells [63,64], as two independent positive controls to validate this assay in *N. vectensis*. Using imaging flow cytometry (ImageStream[X]), we observed morphological differences

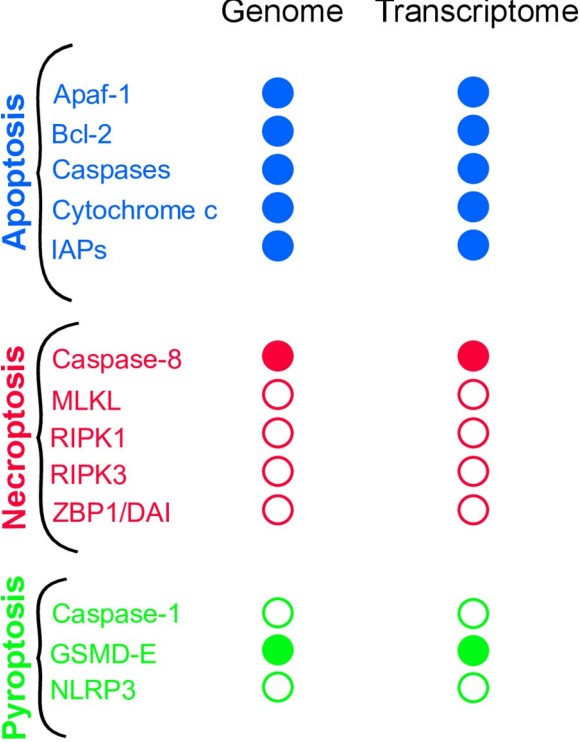

**Fig 2. Presence and expression of *N. vectensis* homologous genes to those involved in the apoptosis, necroptosis, and pyroptosis pathways in mammals.** BLAST searches were conducted on the genome of *N. vectensis* using key genes from the apoptosis, necroptosis, and pyroptosis pathways as queries, complemented by examining the results of the reanalyzed transcriptomic data sourced from Lewandowska *et al.*, 2021 [51]. Filled circles in our results indicate both presence in the genome and expression in the transcriptome, while open circles denote absence in the genome and lack of expression in the transcriptome.

between cells that were negative to Apotracker Green and those that were positive to Apotracker Green and/or the viability dye Zombie NIR (**S1 Fig**). Treatment of *N. vectensis* derived cells with 2 mM of the translation blocker cycloheximide for 48 hours resulted in a significantly increased Apotracker Green median fluorescence intensity (MFI) compared to DMSO (p = 0.001269, n = 3, two-sided t-test) (**S1A and S1B Fig**). MMC, an antitumor drug known to induce apoptosis by crosslinking DNA [64,65], was previously used to kill proliferating cells before candidate stem cell transplantation in *N. vectensis* [66]. Indeed, treatment of *N. vectensis* derived cells with 60 μM MMC for 48 hours resulted in a significant increase in Apotracker Green MFI (p = 0.013360, n = 3, two-sided t-test) (**S2C and S2D Fig**). Notably, cells that were negative to Apotracker Green had a significantly higher circularity score compared to Apotracker Green positive cells, and compared to cells that were positive to both Apotracker and Zombie NIR, the latter had the lowest score (**S1C Fig**). Similarly, zygotes treated with 60 μM MMC for 48 hours had a significantly increased Apotracker Green MFI compared to their DMSO counterparts (p = 0.002472, n = 4, two-sided t-test) (**S2E and S2F Fig**). Next, we studied the effect of poly(I:C) on apoptosis. In mammals, poly(I:C) is recognized, among other intracellular receptors, by the extracellular toll-like receptor 3 (TLR3) which is expressed on the cell surface of fibroblasts and epithelial cells [67]. *N. vectensis* has only one toll-like receptor (TLR), which is not orthologous of TLR3 [68]. This receptor was shown to respond to bacterial pathogens and its exact ligand is still unknown, however, was suggested to be Flagellin [68]. Indeed, injection of poly(I:C) into *N. vectensis* zygotes resulted in strong upregulation of RLRa and RLRb (**S2G Fig**), whereas incubation of zygotes with high concentrations of poly(I:C) (0.5 μg/μL) did not increase the expression of neither RLRa nor RLRb (**S2H Fig**), suggesting that *N. vectensis* can only respond to intracellular dsRNA. To deliver poly(I:C) into cells, we transfected *N. vectensis*-derived cells with 2 μg/ml poly(I:C), a concentration that is commonly used for transfecting human cell lines [69,70]. Indeed, we detected a significantly increased Apotracker Green MFI in poly(I:C) transfected cells relative to their mock transfected counterparts (p = 0.0265, n = 3, one-sided t-test) (**Fig 3A–3C**). Importantly, green autofluorescence in the *N.vectensis* derived cells was negligible relative to the Apotracker Green signal (**S4A Fig**). In addition, the fraction of Apotracker Green[+] SYTOX Blue[+] cells, which represent late apoptosis and/or necrotic cells in a similar Annexin V based assay [71,72], was significantly increased in poly(I:C) transfected cells compared to their mock transfected counterparts (p = 0.039961, n = 3, two-sided t-test) (**Fig 3D**). Notably, the fraction of cells that were Apotracker Green[-] SYTOX Blue[+], which contains cells that have died due to reasons other than apoptosis, did not change significantly upon poly(I:C) transfection (p = 0.445487, n = 3, two-sided t-test) (**Fig 3E**). Imaging cytometry examination of cells from animals injected with poly(I:C) showed morphological alterations characteristic of apoptosis [73]. These changes, including membrane blebbing, the formation of apoptotic bodies, and cell shrinkage, were identified using Apotracker Green staining combined with the viability dye Zombie NIR (**S3 Fig**).

As of today, no cytokines have been described in *N. vectensis*. Thus, it is unclear whether infected cells are capable of alarming neighbors upon viral infection (i.e., paracrine signaling), or alternatively, the signaling is limited to the infected cells (i.e., autocrine signaling). To address this question, we transfected cells with poly(I:C) labeled with rhodamine. Flow cytometry revealed transfection efficiency ranging from about 29% to 44% (**S5A–S5C Fig**). Next, we asked whether cells that were successfully transfected with poly(I:C) (rhodamine positive) were prone to apoptosis more than cells that did not uptake the labeled poly(I:C). Indeed, the Apotracker Green signal was increased in cells that were also positive to rhodamine (p = 0.00081, n = 3, one-sided t-test) (**Fig 3F and 3G**), indicating that poly(I:C) triggered apoptosis mostly in cells that internalized it.

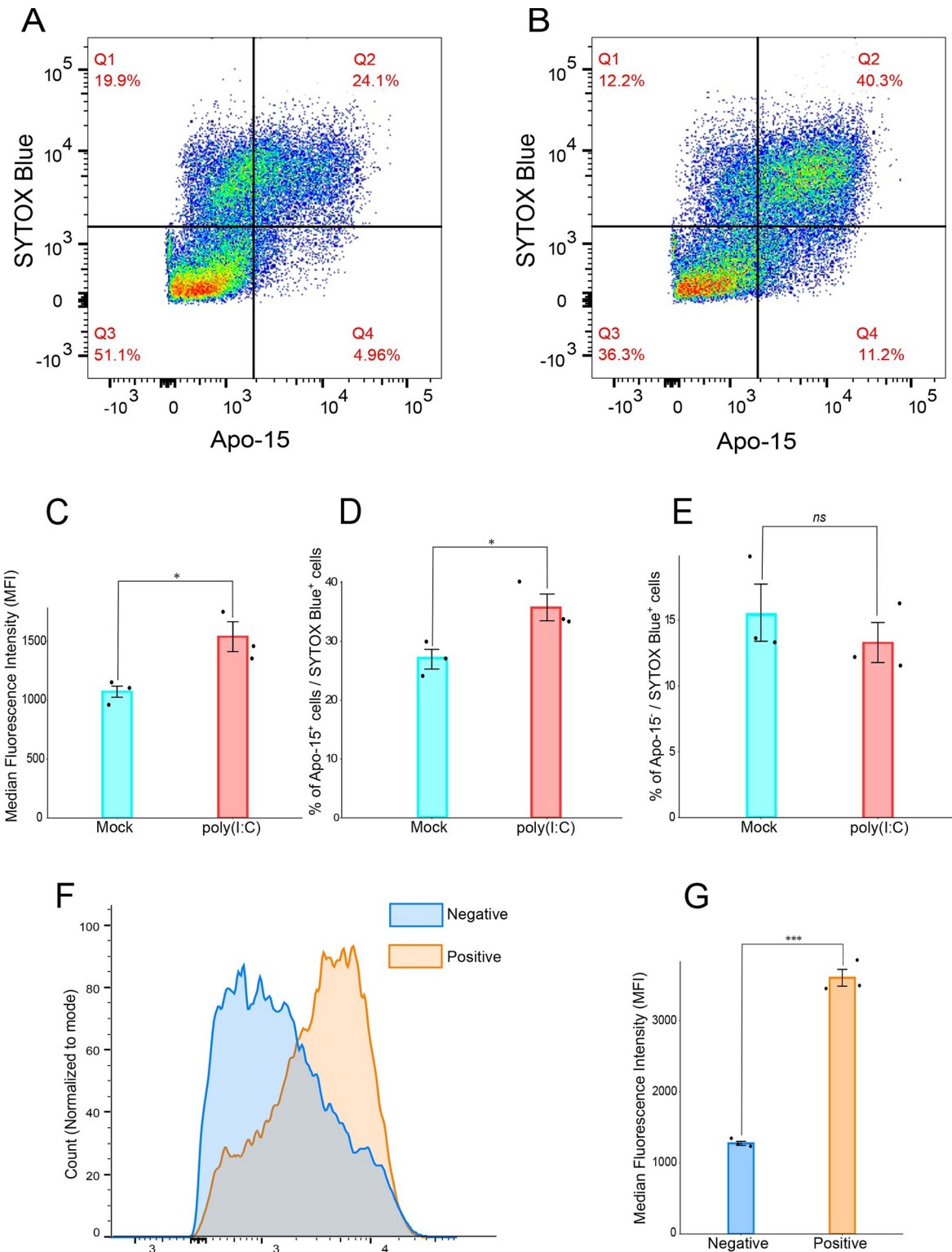

**Fig 3.** poly(I:C) induces cell death *ex vivo*. Apotracker Green was used to quantify apoptosis 24 hours following 2 μg/ml poly(I:C) transfection in *N. vectensis* derived cells. (A) representative flow cytometry result of mock transfected *N. vectensis* derived cells. (B) representative flow cytometry result of poly(I:C) transfected *N. vectensis* derived cells. (C) Quantification of the data (shown in a and b). The bars represent median fluorescence intensity (MFI). (D) Average percentage of Apotracker Green+/sytox-blue+ fraction of cells. (E) Average percentage of Apotracker Green-/sytox-blue+ fraction of cells. (F) *N. vectensis* derived cells were transfected with poly(I:C)

labeled with rhodamine. Representative histogram of Apotracker Green FMI in rhodamine negative cells (shown in blue) versus rhodamine positive cells (shown in orange). (G) Quantification of the data shown in (F). In all graphs, error bars represent standard deviation. Three biological replicates were used for each condition. 30,000 events were recorded per sample. All comparisons were done by one sided t-test. Individual data points are shown as a jitter. * p<0.05, **p<0.01, ***p<0.001.

## dsRNA induces apoptosis in *N. vectensis* injected animals

Caspases are key mediators of apoptosis. Among them, Caspase-3 is a major executioner caspase involved in the execution phase of apoptosis [74]. Following cleavage and activation by initiator caspases (CASP8, CASP9 and/or CASP10), the mammalian Caspase-3 executes apoptosis by catalyzing cleavage of many essential key proteins [74–77]. Methods for determining Caspase-3 activity are based on the proteolytic cleavage of its preferred recognition sequence Asp-Glu-Val-Asp (DEVD) [78]. Such methods have been applied for studying apoptosis upon stress conditions in other cnidarians including corals and the sea anemone *Exaiptasia diaphana* [11,79,80]. To study the effect of dsRNA *in vivo* we first validated this assay by incubating zygotes with 60 µM MMC and compared them to DMSO treated controls. Indeed, 24 hours MMC treatment resulted in about 2-fold increase in Caspase-3 activity (**S6A Fig**), whereas 48 hours incubation resulted in a slightly stronger activity of about 2.75-fold increase (**S6B Fig**). Next, we compared the Caspase-3 activity in poly(I:C) injected animals *versus* NaCl control at different time points. Three batches of zygotes were used as biological replicates for each condition. At 6 hpi, we did not observe a significant difference in Caspase-3 activity compared to NaCl control (p = 0.56, n = 3, one-sided t-test) (**Fig 4A**), whereas caspase activity was significantly increased at 24 hpi (p = 0.0013, n = 3, one-sided t-test) and did not reach a statistical significance at 48 hpi (p = 0.069, n = 3, one-sided t-test) (**Fig 4B and 4C**). The increase in Caspase-3 activity relative to 6 hours was most noticeable at 24 hours (p = 0.145, n = 3, one way ANOVA test) after injection and decreased in magnitude at 48 hpi (p = 0.469, n = 3, one way ANOVA test), however, these differences did not reach statistical significance (**Fig 4D**). In agreement with our transcriptomic analysis (**Fig 1A and 1B**), the response to poly(I:C) was strongest at 24hpi. Altogether, these results indicate that poly(I:C) is sufficient to activate Caspase-3 *in vivo* in a time dependent manner.

## Caspases in Metazoa: Phylogenetic analysis and domain organization

To broaden our understanding of the evolutionary history of caspases in Metazoans, we conducted tBLASTn and BLASTp searches of caspases on NCBI, Uniprot, and on available genomes of Metazoa representing Ctenophora, Porifera, Cnidaria, Protostomia, and Deuterostomia. We identified 160 putative caspases and of these six putative caspase genes were identified in the genome of *N. vectensis* (**S2 Table**). Five phylogenetic analyses were conducted to explore the distribution of homologs of these caspase genes from *N. vectensis* across Metazoa. The Maximum-Likelihood tree depicted in **Fig 5** was inferred from full-length sequence alignment generated by MAFFT. Although the support values are very similar across all analyses, this result displays the highest support values. This phylogenetic tree shows that the caspases from *N. vectensis* belong to five distinct clades, with the exception of clade II (highlighted in green), where Caspase-9 is exclusively found in deuterostome animals. All caspases within this clade possess a CARD and CASc domain; in humans, they are recognized as initiator caspases [81]. The sequences in the tree were named based on their human orthologs. Clade I exclusively comprises sequences from Anthozoa, and showcases a domain organization with both NHR and CASc domains. The Neuralized Homology Repeat (NHR) domain, a module of approximately 160 amino acids, was initially identified as a tandem repeat in the *Drosophila* Neuralized, a protein instrumental in the development of the central and peripheral nervous

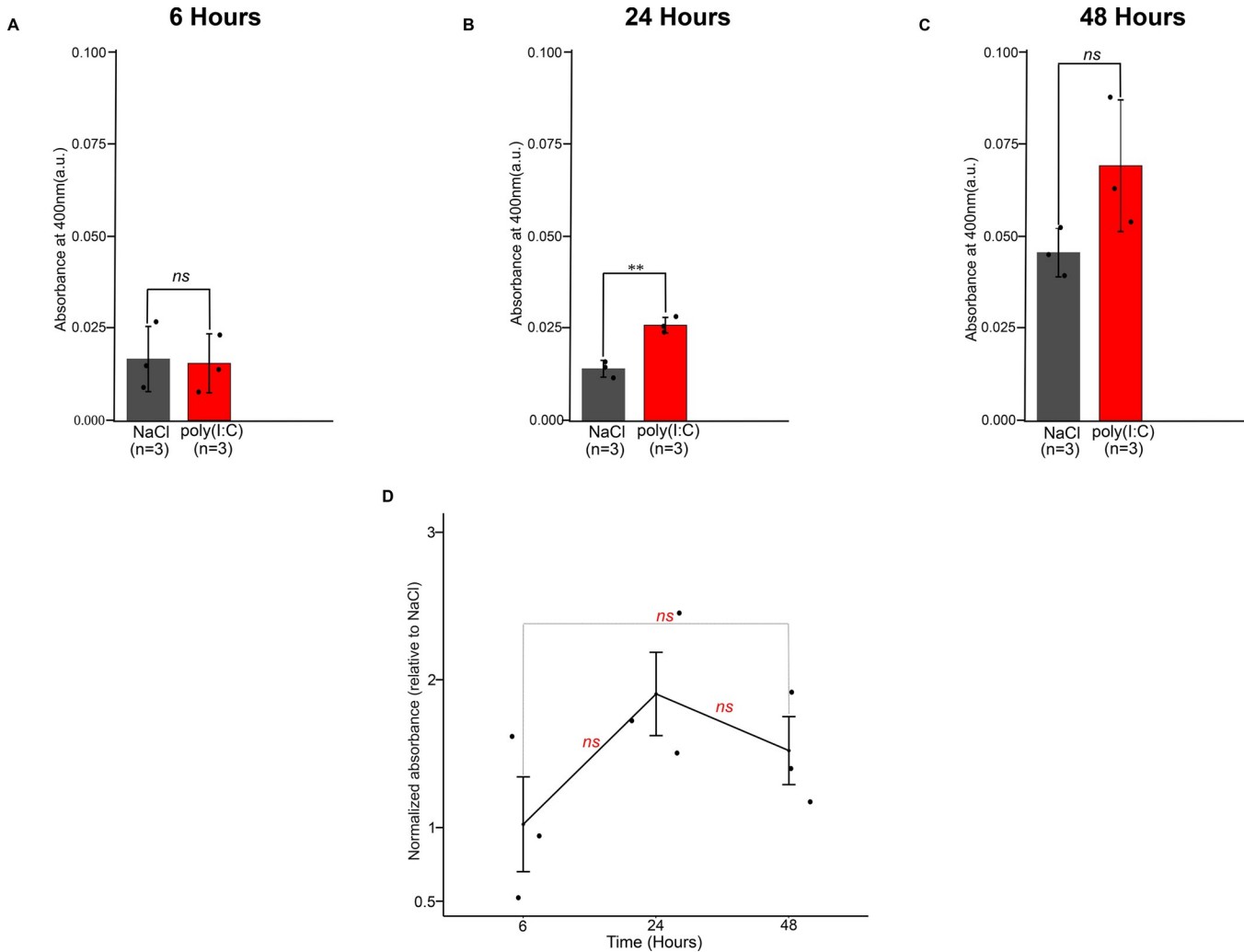

**Fig 4.** poly(I:C) increases caspase-3 activity *in-vivo*. Zygotes were injected with poly(I:C) and subjected to Caspase-3 activity colorimetric assay. Optical density was measured at 400nm. (A) Lysates were obtained from either poly(I:C) or NaCl control treated embryos at 6 hours post injection (hpi), (B) at 24 hpi, and (C) at 48 hpi. (D) Optical density in poly(I:C) treated animals (relative to its NaCl treated controls) at different time points. Three biological replicates each of which consisting of 3 batches of zygotes were used for each condition. For panels (a)-(c) one-sided t test was performed. For panel (D) one way ANOVA test with Tukey's post hoc test was performed. Individual data points are shown as a jitter. * p<0.05, **p<0.01, ***p<0.001.

systems [82]. Intriguingly, within this clade, we discovered a sequence from *N. vectensis* (NVE5282) that is the most upregulated at 24 hpi time point, exhibiting LFC of 6.4. Clade III, highlighted in red, encompasses sequences from Protostomia, Deuterostomia, Medusozoa, and Anthozoa lineages. A defining feature of the caspases in this clade is the presence of dual death-effector domains (DEDs), which serve as recruitment domains, alongside CASc domains. Caspases with this domain structure (Caspase-8 and -10 in humans) are identified as initiator caspases in the apoptotic pathway [83]. Within this clade, a sequence from *N. vectensis* (NVE9681) was found to be upregulated at the 24 hpi time point, with a LFC of 2.8. Clade IV, highlighted in yellow, displays the greatest diversity regarding the number of human orthologs present as well as domain structure. This clade includes sequences from the Protostomia, Deuterostomia, and Anthozoa lineages. Within this clade, we observe subclades wherein inflammatory caspases from humans (Caspase-1, -4 and -5) are located. Adjacent to this subclade,

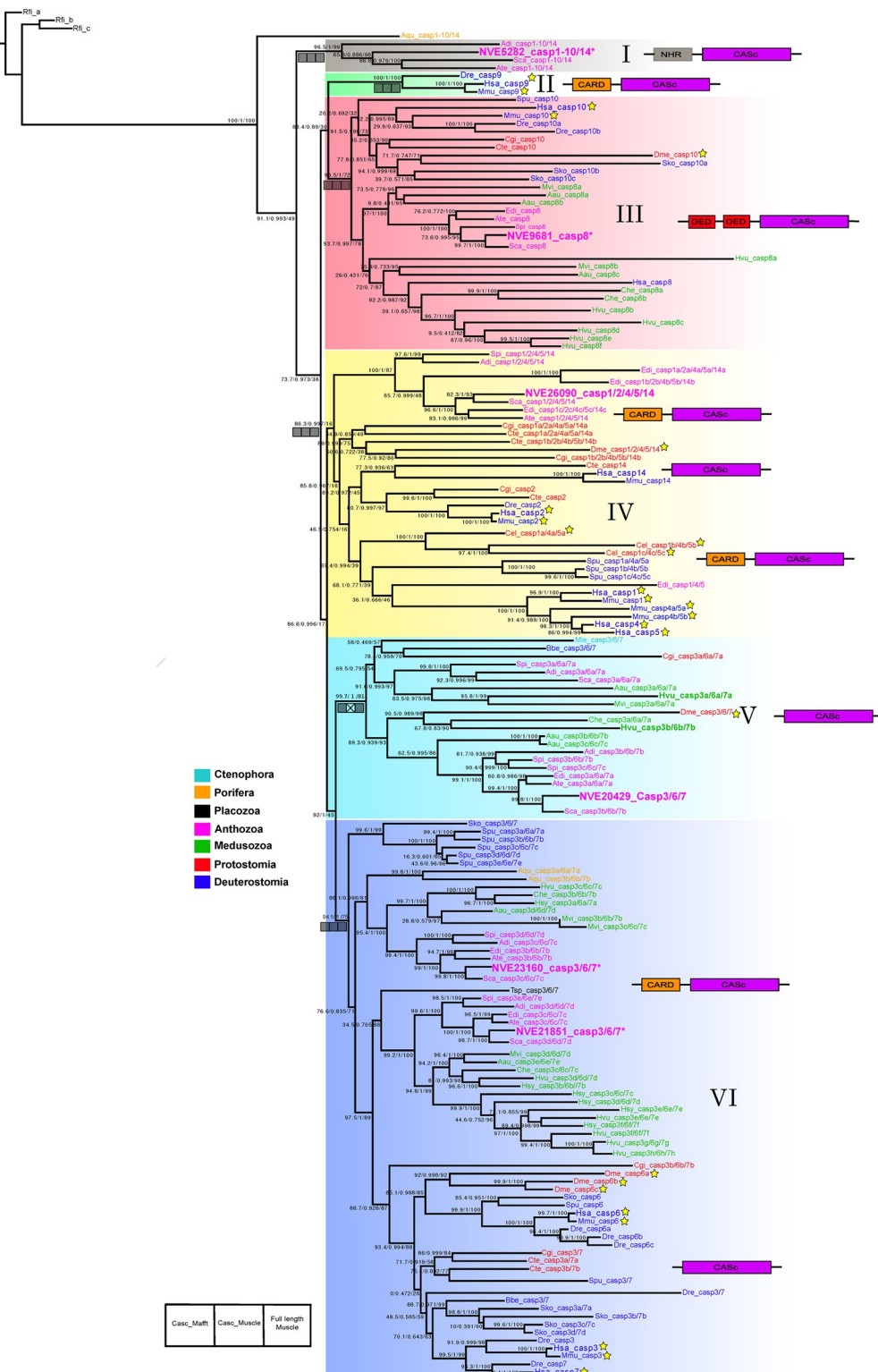

**Fig 5. Phylogenetic relationship among metazoan caspases made with the full-length sequence alignment generated by MAFFT.** Numbers on nodes, separated by slashes, represent SH-aLRT, approximate Bayes test, and ultrafast bootstrap support values, in that order, for Maximum Likelihood (ML) analysis. Navajo's rug below the nodes illustrates the results of the sensitivity analysis (using different alignments) based on maximum likelihood. Grey squares mean the clade is recovered in the analysis, white with an X ones were not. The domain organization is

displayed alongside each clade. CASc: caspase domain, CARD: caspase-recruitment domain, DED: death-effector domain, NHR: neuralized homology repeat. Sequences from *N. vectensis* are highlighted in bold magenta and presented in a larger font. Some of these sequences feature an asterisk, indicating that these genes are upregulated at the 24hpi time point with a Log2 Fold Change > 1. A yellow star placed in front of certain branches denotes well-studied species (Cel: *Caenorhabditis elegans*, Dme:*Drosophila melanogaster*, Hsa: *Homo sapiens* and *Mmu*: *Mus musculus*). Rfi: *Reticulomyxa filosa*, Aqu: *Amphimedon queenslandica*, Adi: *Acropora digitifera*, NVE: *N. vectensis vectensis*, Sca: *Scolanthus callimorphis*, Ate: *Actinia tenebrosa*, Dme: *Drosophila melanogaster*, Dre: *Danio rerio* Hsa: *Homo sapiens*, Mmu: *Mus musculus*, Spu: *Strongylocentrotus purpuratus*, Cgi: *Crassostrea gigas*, Cte: *Capitella teleta*, Sko: *Saccoglossus kowalevskii*, Mvi: *Morbakka virulenta*, Aau: *Aurelia aurita*, Edi: *Exaiptasia diaphana*, Spi: *Stylophora pistillata*, Hvu: *Hydra vulgaris*, Che: *Clytia hemisphaerica*, Cel: *Caenorhabditis elegans*, Mle: *Mnemiopsis leidyi*, Bbe: *Branchiostoma belcheri*, Tsp: *Trichoplax sp*., Hsy: *Hydractinia symbiolongicarpus*.

human Caspase-2, identified as an initiator caspase, along with Caspase-14, recognized for its functional role in development, are also located [84]. In this clade, there is a sister group to the remaining species, exclusively consisting of anthozoan sequences, which possess both CARD and CASc domains. Within this group, a sequence from *N. vectensis* (NVE26090) is found, however, it is not upregulated at any time point in our experiment. Clade V stands as the only group devoid of human orthologs, predominantly featuring sequences from Anthozoa and Medusozoa. The protein domain structure within this clade is characterized by the sole presence of the CASc domain. Within this clade, two sequences from *Hydra vulgaris* appear in bold. Recently, these sequences were discovered to have an inflammatory function [85]. Within this clade, a sequence from *N. vectensis* (NVE20429) is found, which is not upregulated at any time point in our experiment. Clade VI represents the largest clade both in terms of the number of sequences and the diversity of lineages. It encompasses sequences from Porifera, Placozoa, Protostomia, Deuterostomia, Medusozoa, and Anthozoa. In this clade, we identified a subclade whose sequences are characterized solely by the presence of the CASc domain, and exclusively include sequences from the Protostomia and Deuterostomia lineages. The human Caspase-3, -6, and -7, grouped within this subclade, are identified as executioner caspases in the apoptotic pathway [84]. The remaining subclades, apart from possessing the CASc domain, also feature an N-terminal CARD domain. Within these subclades, we identified two upregulated sequences from *N. vectensis*: NVE23160 with a LFC of 1.15 and NVE21851 with a LFC of 3.37 at 24 hpi. It is noteworthy that the tree we obtained using Bayesian inference is similar to this obtained through the maximum likelihood method (**S7 Fig**).

Although the ultrafast bootstrap values supporting the six delineated clades (I-VI) exceed 95 only for clades I and II, the support for the remaining four clades ranges from 16 to 81. Notably, clade IV is the only one with a bootstrap value below 70. Despite this, we obtain high support from both the SH-aLRT and the Bayes test for these clades (**Fig 5**). This discrepancy could arise from a matter of statistical sensitivity, wherein SH-aLRT and aBayes tests may be more sensitive to certain data patterns. They might detect signals in the data supporting the branch that bootstrap analysis might miss, especially if the dataset is small or the signal is weak [86]. Furthermore, it is important to note that the results of the five phylogenetic analyses, including two phylogenetic analyses based on only the Casc domains, and three analyses based on full length sequence were highly congruent regarding the tree structure and particularly in identifying the six clades delineated here (I-VI), demonstrating the robustness of the results (**Fig 5** and **S7A–S7D**). This is illustrated in **Fig 5** using navajo rugs representing the results of the sensitivity analysis.

## Phylogenetic reconstruction of two families involved in the apoptosis pathway

Bcl-2 and Apaf-1 protein sequences were retrieved from Uniprot, NCBI, and downloaded genomes. This was conducted using the methodology described below (see *Methods*) for the

caspase family. The two topologies obtained are shown (**S9A and S9B Fig**). Nine putative Bcl-2 proteins were identified in *N. vectensis* and were dispersed throughout the phylogeny (**S9A Fig**). Five of these proteins clustered with representatives from the anti-apoptotic functional group, where two distinct groups, Bcl-2 and Bcl-b, were clearly identified. Additionally, three putative Bcl-2 proteins of *N. vectensis* clustered within the pro-apoptotic clades, Bax and Bok. Three putative Apaf-1 proteins were identified in *N. vectensis* (**S9B Fig**). NVE2160, the gene upregulated in our study, is clustered in a clade characterized by sequences that possess the CARD and NB-ARC domains, receiving the highest support. These results are similar to the phylogenies previously obtained in other studies for these two protein families [87,88].

### dsRNA induces a conserved antiviral response pathway across metazoans

To obtain a wider picture on the evolution of the antiviral apoptotic response in animals, we analyzed the results from seven studies [51,89–94] focused on the gene expression response to immune challenges (**Fig 6A**). Most of these studies specifically employed poly (I:C) as a viral mimic. Importantly, Bar Yaacov's 2022 study [89] on the planarian *Schmidtea mediterranea* employed a double-stranded RNA virus, rather than poly (I:C), as the immunological challenge. Despite differences in the species studied, experimental designs, and analytical methods, these studies collectively contribute to our understanding of dsRNA's influence on apoptosis on an evolutionary scale. Based on the results of the gene ontology (GO) analysis conducted in each study, we found that GO terms related to apoptosis are significantly enriched in all the studies, except for the hemichordate *Saccoglossus kowalevskii*, which did not show any significant GO terms related to apoptosis (**Fig 6B** and **S1 Text**). Similarly, under the apoptosis categories, we pinpointed clusters of upregulated genes within crucial protein families in the apoptotic pathway, including caspases (both initiators and executioners), anti-apoptotic regulators like the Bcl-2 family members, and pro-apoptotic factors such as Apoptotic Protease-Activating Factor 1 (Apaf-1), Tumor Necrosis Factor (TNFs), and TNF Receptor Associated Factor (TRAFs) (**Fig 6B**). The majority of lineages exhibited a shared upregulation of these genes.

### Discussion

Our study offers an extensive phylogenetic examination of the caspase protein family. Recently, another study was published, providing a comprehensive phylogenetic analysis of the same family [95]. This study presents certain parallels with our study, especially in the identification of specific monophyletic groups. For instance, both analyses recognize a group, which we term Group III, characterized by its dual Death Effector Domains (DED). Nonetheless, there are significant differences as well. While the Krasovec et al [87] study recovers caspase groups based on domain structure, our topology diverges from this pattern. This discrepancy likely stems from differing taxonomic sampling strategies. Our analysis, in contrast to Krasovec et al. [87] which uses 76 sequences, incorporates 163 sequences from various major metazoan lineages, with a particular emphasis on the Cnidaria phylum. These methodological differences highlight the distinct perspectives each study brings to understanding the evolution of the caspase family. Additionally, our phylogenetic analysis primarily aims to ascertain the placement of different *N. vectensis* genes within this evolutionary context. Contrasting with Krasovec's et al. findings [87], the 2016 study by Moya et al. [88], which features a comprehensive phylogenetic analysis of the caspase family, is highly congruent with our conclusions. For instance, their analysis places the NEMV 201 sequence, equivalent to our NVE9681_casp8 sequence, closer to the clade they identify as 'Vertebrate caspase-8-10'. Similarly, the NEMV205 sequence, corresponding to our NVE26090_casp1/2/4/5/14 sequence, is

A

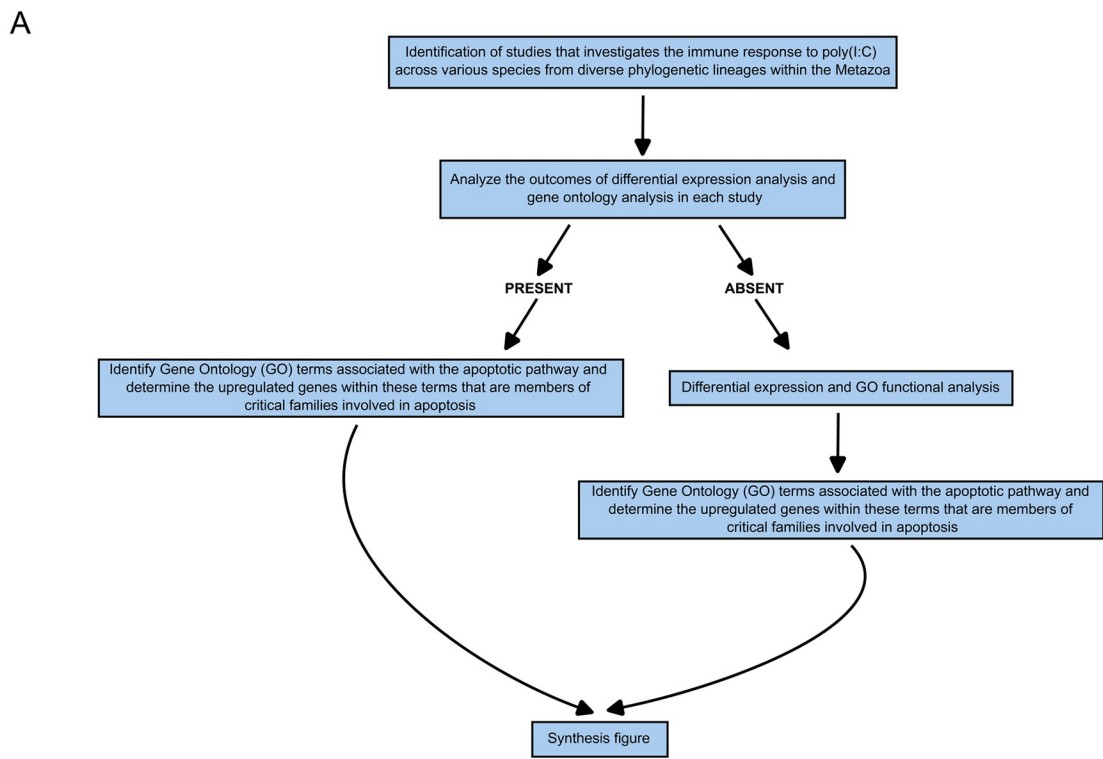

B

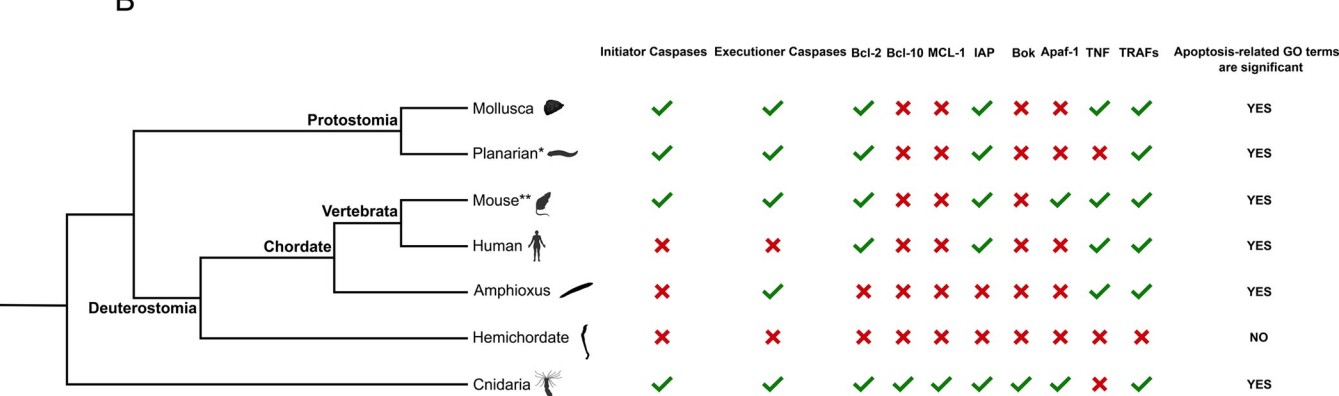

**Fig 6. Synthesis on the evolution of the antiviral apoptotic response in animals.** (A) Flowchart depicting the steps to gather information on the studies of interest. (B) A simplified phylogenetic tree representing selected groups, highlighting the regulatory patterns of key apoptosis gene families. In this representation, a check mark (√) denotes the upregulation of a gene, while an X indicates the absence of upregulation following exposure to the dsRNA viral mimic poly(I:C) or virus. *The study pertaining to Platyhelminthes utilized a dsRNA virus as opposed to poly(I:C). **This information is synthesized from two separate studies. The specific details of each study are available in **S1 Text**.

also situated in the same clade, closely related to human caspase 2 and 14. This similarity may stem from Moya and collaborators' [88] extensive sampling within the Cnidaria phylum, akin to the methodology of our study.

Utilizing these phylogenetic results along with insights gleaned from the literature [87], we reconstructed the distribution of various caspase types across Metazoa (**S8 Fig**). The evolutionary history of caspases in Metazoa is particularly noteworthy, as all metazoan groups share one

specific type: executioner caspases. Additionally, the frequent losses observed in the phylum Nematoda highlight a complex evolutionary trajectory for this protein family, characterized by numerous losses, acquisitions, and diversifications.

The co-evolution of viruses with their hosts has led to the emergence of viral pathogens that are capable of evading or actively suppressing host immunity [96]. The ability to recognize and respond to foreign nucleic acids among an abundance of self nucleic acids in an organism is, therefore, crucial for host fitness [97]. Long dsRNA in the cytosol is a hallmark of DNA and RNA virus replication and is absent from an uninfected host cell [33,98]. In vertebrates, poly(I: C), which mimics dsRNA, binds to and triggers the activation of the RNA sensors PKR, OAS1, TLR3, MDA5 and RIG-I, and consequently poly(I:C) or its derivatives have been used as a tool to identify and characterize dsRNA recognition receptors and ligand requirements [38,99–103]. Following recognition of poly(I:C), MDA5 signals via MAVS and IRF3-IRF7 to induce type I INFs which enables infected cells to "alert" neighboring cells against incoming infection and recruit cells of the immune system to battle the virus [104,105]. Additionally, MAVS activation promotes apoptosis through several pathways [106–110].

In contrast to vertebrates, most invertebrates, such as insects and nematodes, lack interferons and are thought to rely mainly on RNA interference (RNAi) for antiviral defense [97]. However, limited phyletic sampling makes it impossible to determine what was the original mode of action of these systems in their last common ancestor [111] and whether apoptosis was also induced by viral challenges in early animals. We have previously shown that *N. vectensis* responds to poly(I:C) by upregulating immune-related genes [51] but it remained unclear whether the observed gene expression changes are associated with a phenotype. Here, we filled that gap by showing that *N. vectensis* is capable of inducing apoptosis by exposure to dsRNA alone, both *ex-vivo* (**Fig 3**) and *in-vivo* (**Fig 4**). In fact, regulation of apoptosis was the most significantly enriched set of genes in GO analysis (**Fig 1B**). Our phylogenetic analysis (**Fig 5**) revealed three orthologs that were upregulated following poly(I:C) treatment, corresponding to the two types of caspases (initiator and executioner) involved in mammalian apoptosis. NVE9681_casp8 belongs to clade III (**Fig 5**), which comprises exclusively initiator caspases, and was found to be upregulated at the 24 hpi time point with a LFC of 2.8, NVE23160_casp3/6/7 with a LFC of 1.15, and NVE21851_casp3/6/7 with a LFC of 3.37 at 24 hpi, both were identified in the clade VI (**Fig 5**), which comprises executioner caspases.

Our results challenge the established dichotomy in the literature [111] regarding the antiviral immune systems of vertebrates, which predominantly depend on the interferon pathway, and invertebrates, which are believed to primarily rely on RNAi. It appears that *N. vectensis* can activate apoptosis similarly to vertebrates in response to dsRNA challenge. Our transcriptomic analysis (**Fig 1**) indicated that both intrinsic and extrinsic pathways are activated in response to dsRNA. This result is in agreement with our cross species comparative transcriptomic analysis (**Fig 6**) that demonstrated that: (1) most representative metazoans (6 out 7 analyzed) respond to dsRNA by significantly upregulating genes related to apoptosis, and (2) the upregulated genes were mostly conserved between distant species. However, in some lineages it seems that the apoptotic response to dsRNA was lost. For instance, the hemichordate *Saccoglossus kowalevskii* did not exhibit a transcriptomic response to poly(I:C) [112]. Similarly, in the nematode C. *elegans*, which has been extensively studied in the context of apoptosis, dsRNA injection did not induce apoptosis [113]. It is possible that apoptosis was decoupled from dsRNA in C. *elegans* as a consequence of a highly efficient RNAi response.

In recent years, a growing body of evidence indicate that other forms of cell death are also involved in the antiviral response. In particular, necroptosis, a regulated form of necrotic cell death, and pyroptosis, a regulated form of inflammatory lytic cell death mediated by gasdermin proteins [55], are important host defense strategies for eliminating viral infected cells. The

latter was observed in Cnidaria in the context of the antibacterial response [45,46]. Our phylogenetic analysis revealed that the *N. vectensis* Caspase gene (NVE20429) orthologous to the *Hydra vulgaris* gene (**Fig 5**) which was recently reported to be involved in pyroptosis was not upregulated upon poly(I:C) treatment. However, a Gasdermin-encoding gene (NVE20414) was upregulated (LFC = 1.8, p = 0.0014) at 24 hpi. Importantly, gasdermins are regulated post-translationally by proteolytic cleavage [114]. Therefore, based on our data we cannot exclude the possibility that pyroptosis is also activated in response to dsRNA in *N. vectensis* (**Fig 2**). Further research is required to determine whether pyroptosis and apoptosis are activated simultaneously or whether these two processes are mutually exclusive in *N. vectensis.*

In conclusion, our findings in *N. vectensis*, together with our cross-species analysis in the current study, suggest that induction of apoptosis by dsRNA is an ancient response that was present already in the last common ancestor of cnidarian and bilaterian animals, that lived 600 million years ago.

## Methods

### Sea anemone culture

*N. vectensis* polyps were grown in 16 ‰ sea salt water (Red Sea, Israel) at 18˚C. Polyps were fed with *Artemia salina* nauplii three times a week. Spawning of gametes and fertilization were performed according to a published protocol as follows: in brief, temperature was raised to 25˚C for 9 h and the animals were exposed to strong white light. Three hours after the induction, oocytes were mixed with sperm to allow fertilization.

### Cell dissociation

Animals were dissociated into single cells as previously described [115]. Briefly, planulae were washed twice with calcium/magnesium free and EDTA free artificial seawater (×3 stock solution is: 17 mM Tris-HCl, 165 mM NaCl, 3.3 mM KCl and 9 mM of NaHCO$_3$; final solution pH 8.0) and incubated with 50 μg/ml liberase TM (Roche, Switzerland) at room temperature for 5–10 minutes with occasional gentle pipetting, until fully dissociated. The reaction was stopped by adding 1/10 volume of 0.5 M EDTA solution. The suspension was filtered using a 35 μm cell strainer. Cells were then centrifuged at 500 × *g* at 4˚C and resuspended in 1× calcium/magnesium free sterile PBS (Hylab, Israel). Cells were counted on hemocytometer and viability was determined using trypan blue (Thermo Fisher Scientific, USA).

### Primary cell culture

Animals were grown at 25˚C for 48 hours post fertilization and dissociated into single cells as described in the previous section. Cells were washed 3 times in PBS before seeding. Viability was assessed using trypan blue. Cell suspension with viability of more than 80% (ranging from 81.7% to 91.1%) was used for cell culture. 200,000 cells were seeded per well in a 24 well plate. Cells were grown at 25˚C in L-15 medium (Biological Industries, Israel) supplemented with 10% FBS, 1% Pen-Strep, and 2% 1M HEPES buffer solution.

### Transfection

Transfection was done using lipofectamine 3000 (Thermo Fisher Scientific) according to the manufacturer's instructions, using high dose lipofectamine. A control was implemented by conducting mock transfection, using the same lipofectamine formulation as the poly(I:C) transfection but excluding poly(I:C).

## Flow cytometry

FACSAria III (BD Biosciences, USA) equipped with 405 nm, 408 nm, 561 nm and 633 nm lasers was used to quantitatively assess apoptosis. Per run, 30,000 events were recorded. Helix NP (SYTOX Blue) (Ex430/Em470) (Biologend, USA) was used to determine viability. To monitor apoptosis, Apotracker Green (Ex488/Em520) (BioLegend, USA) was used according to the manufacturer's instructions. FCS files were further analyzed using FlowJo V10 (BD Biosciences, USA). Each measurement consisted of 3 biological replicates, unless indicated otherwise.

## Imaging cytometry

Dissociated cells from zygotes injected with poly(I:C) or treated with either CHX or DMSO control were assessed quantitatively using an Amnis ImageStream$^X$ Mk II apparatus (Luminex, USA) equipped with 405nm, 488nm, 642nm and 785nm (SSC) lasers and 6 acquisition channels, using a 60X magnification objective, with low flow rate/ high sensitivity using INSPIRE200 software (Luminex, USA). The INSPIRE200 software was set up using the following parameters: Channel 01 (bright field), Channel 02 for detecting laser 488nm, and Channel 05 detecting laser 642nm. Dissociated cells were stained with 200nM Apotracker Green (Ex488/Em520) (Biolegend) and with 1:100 dilution of the Zombie NIR (Ex642/Em746) fixable viability dye (Biolegend) for 15 minutes at room temperature. Cells were then washed twice with PBS and resuspended in 100 μL of PBS in 1.5 ml Eppendorf tubes. About $10^7$ cells per sample were used for data acquisition. A total of 10,000 single events were acquired within the focused singlet gate. Focused events were determined by the Gradient RMS parameter for Ch01 and single cells were determined by plotting the area of the events (X-Axis) vs. aspect ratio (Y-Axis) for Ch01. All subsequent analysis was done on this population of cells. Image analysis was run using IDEAS 6.3 software (Luminex, USA), circularity feature was determined using the Shape Change Wizard, which provided circularity score for each population by measuring the degree of the mask's deviation from a circle.

## Caspase3 activity assay

For each condition (NaCl vs. poly(I:C)), about 150 injected animals were snap frozen in liquid nitrogen and stored in -80˚C. Caspase-3 Assay Kit (colorimetric) (Abcam, UK) was used according to the manufacturer's instructions. Briefly, cell pellet was lysed in lysis buffer using a homogenizer. Protein quantification was done using BCA assay (Cyanagen, Italy). Same amounts of protein were loaded into each reaction, ranging from 30 μg to 40 μg depending on the yield. Absorbance was measured at 400 nm after 2 hours using a microplate reader (Bio-Tek, USA). For the experiments shown in **S6A and S6B Fig** Caspase-3 Assay Kit (colorimetric) (Abbkine, USA) was used according to the manufacturer's instructions. For these experiments, protein quantification was done by measuring absorbance at 280nm using Epoch 2 microplate reader equipped with Take3 Plate (BioTek, USA).

## Poly(I:C) microinjection

To stimulate the antiviral immune response in *N. vectensis*, we used poly(I:C) (Invivogen, USA) as dsRNA viral mimic. We used 3.125 ng/ml of high molecular weight (HMW) poly(I:C) in 0.9% NaCl (with an average size of 1.5–8 kb), and 0.9% NaCl as a control. This concentration was determined as sub-lethal in previous titration experiments [51]. In each experiment 100–150 zygotes were microinjected and flash frozen in liquid nitrogen at 6 hours, 24 hours, and 48 hours after injection, and then stored at -80˚C until processed.

## Mitomycin-c and cycloheximide treatments

Zygotes were de-jellified using 3% cysteine (Merck Millipore, USA), washed 6 times in *N. vectensis* medium (16 ‰ artificial sea water), and incubated with either 60 μM mitomycin-c (Sigma-Aldrich, USA), 2 mM cycloheximide (Abcam), or an equivalent concentration of DMSO in 24-well plate containing 500 μL *N. vectensis* medium. 200–500 zygotes were grown per well. 48 hours later cells were dissociated from planulae as described in previous sections, stained with Apotracker Green and subjected to flow cytometry. For *ex vivo* experiments, cells were dissociated and seeded in 24-well plates as described in *Primary cell culture* section. 2 mM cycloheximide (Abcam) or an equivalent concentration of DMSO (Sigma-Aldrich) was used to treat dissociated cells for 48 hours. Cells were then subjected to flow cytometry as described in previous sections.

## RT-qPCR

To test the effect of extracellular poly(I:C), 50 zygotes were incubated with 0.5 μg/μL of poly(I:C) in a total volume of 1 ml 16 ‰ sea salt water (growth medium) (Red Sea, Israel) for 24 hours at 24˚C. as controls, 50 zygotes were incubated only with growth medium. The animals were harvested and snap frozen in liquid nitrogen 24 hours post fertilization. We compared this result to poly(I:C) injection as follows: 200 zygotes were injected with 3.125 ng/μL of poly(I:C) and snap frozen at 24 hpi. For the control group, 200 zygotes were injected with 0.9% NaCl. Total RNA was extracted using TRIzol reagent (Thermo Fisher Scientific) according to the manufacturer's instructions. 500 ng of RNA was used as a template for cDNA synthesis using super script III reverse transcriptase (Thermo Fisher Scientific). Fast SYBR Green Master Mix (Thermo Fisher Scientific) was used for amplifying the cDNA. The samples were measured using the StepOnePlus Real-Time PCR System v2.3 (ABI, Thermo Fisher Scientific). Primer sequences are available in **S3 Table**.

## GO functional analysis

Raw reads files obtained from Lewandowska et al. 2021 [51] were aligned to *N. vectensis* reference genome (NCBI accession number: GCA_000209225.1) using STAR version 2.7.10a [116] and reads mapped to gene models were summarized using featureCounts Version 2.0.1 [117]. Differentially expressed genes between each treatment and control samples were identified using DESeq2 [118] and edgeR [119] (absolute log2 fold change $> 1$ and adjusted $p < 0.05$). Only genes identified as differentially expressed by both the edgeR and DESeq2 methods were used in subsequent analyses (**S1 Table**). We utilized blastp [120] and DIAMOND [121] for functional annotation to identify the most similar homologs for each protein within the merged *N. vectensis* predicted proteome (comprising both JGI and Vienna annotations). These searches were performed against Uniprot database [122]. Searching for ontology terms was carried out through QuickGO [123]. GO enrichment analysis was performed on sets of upregulated genes using clusterProfiler [124]. The barplot per time point is displaying up to 15 of the most significant GO terms (adjusted $p < 0.05$).

## Phylogenetic analysis

**Sequence dataset construction.** Putative metazoan caspases were discovered using tBLASTn and BLASTp searches with eleven human caspases sourced from Uniprot as initial queries. These queries were then used on NCBI, Uniprot, and downloaded genomes. Subsequently, reciprocal BLAST was conducted, and the query was expanded based on the obtained results. Sequences with an e-value inferior to 1e-5 were retained. We conducted BLAST searches on the genomes of various species, including one Ctenophora, one sponge, nine cnidarians (five anthozoans and four medusozoans), two protostomes and two deuterostomes.

All identified sequences were analyzed with SMART [125] and InterProScan (EMBL-EBI) to verify the presence of caspase (CASc) domains. A full list of all protein sequences used in the BLAST search and phylogenetic analysis are provided in **S2 Table**. On average, we identified six homologs in the genome of each species. *Reticulomyxa filosa* (Foraminifiera) caspase homologues were used as outgroup. Protein sequence alignments were generated for full-length sequence (all domains) and only for caspase domain (Casc) using the L-INS-i algorithm in MAFFT software version 7 [126], with default parameters. In addition, we generated multiple alignments using the MUSCLE software [127] with default parameters, both for the full sequence and only for the caspase domain (Casc). Four final multiple alignments were generated to assess the topological congruence across different alignments.

In addition to the caspase family, we also conducted a phylogenetic reconstruction of two protein families belonging to the apoptotic pathway that were found to be upregulated in our data: apoptotic protease activating factor-1 (Apaf-1) and B-cell lymphoma 2 (Bcl-2). A full list of all protein sequences used for the phylogenetic analysis is provided in **S2 Table**. Protein sequence alignments for these two protein families were generated for full-length sequence (all domains) using the L-INS-i algorithm in MAFFT software version 7 [126].

### Tree reconstruction

Phylogenetic analyses for the four multiple alignments previously generated were conducted using amino-acid alignment (**S1 Data**), employing the Maximum-Likelihood (ML) method in IQ-TREE2 [128] and Bayesian Inference (BI) analysis conducted in MrBayes version 3.2.6 [129], respectively. Models of protein sequence evolution for ML (AIC criteria) and BI (BIC criteria) analyses were estimated using ModelFinder [130]. Branches support of the ML tree was assessed through 1000 ultrafast bootstrapping replicates (UBS) [131], 1000 bootstrap replicates for SH-aLRT [86] and an approximate aBayes test [132]. For the bayesian inference analysis, we used only the final multiple alignment of the full sequence, which was generated by MAFFT. Two independent runs were carried out, each with four simultaneous Markov chains over 15,000,000 generations and sampled every 1000 generations. Convergence was checked using Tracer [133]. The consensus tree, along with the posterior probabilities of clades, was derived from trees sampled across the two runs. 25% of the sampled trees were discarded as burn-in. The resulting trees were visualized using figtree version 1.4.4 (http://tree.bio.ed.ac.uk/software/figtree/). Phylogenetic analyses for the two protein families (Apaf-1 and Bcl-2) were conducted using amino-acid alignment, employing the Maximum-Likelihood (ML) method in IQ-TREE2.

### Cross-species comparative transcriptomic analysis

To contextualize our findings within a comparative evolutionary framework, we analyzed the transcriptomic response to dsRNA viral mimic poly(I:C) across seven species (*N. vectensis*, *Homo sapiens*, *Mus musculus*, *S. kowalevskii*, *Crassostrea gigas*, *Branchiostoma belcheri*, and *Schmidtea mediterranea*) from diverse metazoan phylogenetic lineages [51,89,90,94,112,115,134,135]. Our analysis drew on results obtained from each corresponding study. The aim was to ascertain whether dsRNA impacts apoptosis by examining the outcomes of the gene ontology (GO) analysis conducted in each study, and consequently the outcomes of differential expression analysis conducted in each study, to identify if the genes within the apoptosis pathway are upregulated. Detailed information on each of the studies used is found in **S1 Text**.

### Supporting information

**S1 Fig. ImageStreamX morphological analysis of cycloheximide treated *N. vectensis* derived cells.** *N. vectensis* zygotes were treated with 2 mM cycloheximide (CHX) or an

equivalent concentration of DMSO for 48 hours. Cells were dissociated and stained with Apotracker Green and the viability dye Zombie NIR. (A) Representative images of cells that were negative to Apotracker Green and Zombie NIR (first 3 rows), cells that were positive to Apotracker Green (rows 4–6), and cells that were positive to both Apotracker Green and Zombie NIR (last 3 rows). (B) Histogram comparing the difference in Apotracker Green intensity in DMSO versus CHX treated cells. The summary statistics is shown in the table. (C) Comparison of circularity score across cells that are negative to both markers (live), positive to Apotracker Green (early apoptosis), and positive to both markers (late apoptosis). One way ANOVA test with Tukey's post hoc test was performed. Individual data points (single cells) are shown as a jitter. * $p < 0.05$, **$p < 0.01$, ***$p < 0.001$.
(TIF)

**S2 Fig. Cycloheximide (CHX) and mitomycin-c (MMC) induce apoptosis in *N. vectensis* cells that can be detected using Apotracker Green.** (A) Cells were dissociated from 48 hours old planulae and treated with 2mM cycloheximide or DMSO for 48 hours. Cells were than stained with Apotracker Green and subjected to flow cytometry analysis. Representative result is shown. (B) quantification of (A). Three biological replicates were used. Error bars indicate standard deviation. Cells were dissociated from 48 hours old planulae and treated with 60uM MMC or DMSO control. (C) Representative histogram showing the difference between DMSO and MMC treated cells. (D) Median fluorescence intensity (MFI) in DMSO versus MMC treated cells. (E) Zygotes were incubated with mitomycin-c or DMSO for 48 hours. Cells were then dissociated and subjected to flow cytometry analysis. Representative result is shown. (F) quantification of (E). Four biological replicates were used. (G) Zygotes were injected with poly(I:C) and the mRNA expression levels of RLRa and RLRb were measured by RT-qPCR. (H) Zygotes were incubated with a high concentration of poly(I:C) (0.5 ug/uL) and subjected to RT-qPCR as described in (G). Error bars indicate standard deviation. All comparisons were done by two-sided t-test. Individual data points are shown as a jitter. * $p < 0.05$, **$p < 0.01$, ***$p < 0.001$.
(TIF)

**S3 Fig. ImageStreamX morphological analysis of cells derived from poly(I:C) injected *N. vectensis* zygotes 24 hpi.** Cells were dissociated and stained with Apotracker Green and the viability dye Zombie NIR. (A) Representative images of cells that were negative to both Apotracker Green and Zombie NIR (first 4 rows), cells that were positive to Apotracker Green and negative to zombie NIR (rows 5–8), cells that were positive to Zombie NIR and negative to Apotracker Green (rows 9–12), and cells that were positive to both Apotracker Green and Zombie NIR (rows 13–16). (B) Cells were gated based on their signal to Apotracker Green (X axis) and Zombie NIR (Y axis). Percentage for each quadrant is shown. Q1: Apotracker Green negative/ Zombie NIR positive, Q2: Apotracker Green positive/ Zombie NIR positive, Q3: Apotracker Green negative/ Zombie NIR negative, Q4: Apotracker Green positive/ Zombie NIR negative (C) Comparison of circularity scores across cells that are (left to right): positive to Zombie NIR only (dead/necrotic) (Q1), positive to Apotracker Green only (early apoptosis) (Q4), positive for both markers (late apoptosis/ and/or necrosis) (Q2), negative to both markers (Q3) (live). (D) Area of cells belonging to each population as described in (C). One way ANOVA test with Tukey's post hoc test was performed. Individual data points (single cells) are shown as a jitter. * $p < 0.05$, **$p < 0.01$, ***$p < 0.001$.
(TIF)

**S4 Fig. Green autofluorescence is negligible relative to the Apotracker Green signal.** (A) a histogram showing the green fluorescent signal of SYTOX Blue single stained mock

transfected cells (purple line), mock transfected cells stained with both SYTOX Blue and Apotracker Green (cyan), and poly(I:C) transfected cells stained with both SYTOX Blue and Apotracker Green (red).
(TIF)

**S5 Fig. Transfection efficiency of rhodamine labeled poly(I:C) in *N. vectensis* derived cells.** Dissociated cells were seeded overnight and transfected using lipofectamine 3000. Transfection efficiency was determined after 24 hours by flow cytometry analysis. (A) representative result of cells transfected with 2 μg/ml unlabeled poly(I:C). (B) representative result of cells transfected with 2 μg/ml rhodamine labeled poly(I:C). (C) Quantification of (A) and (B). Error bars indicate standard deviation. 30,000 events were acquired for each replicate. 3 biological replicates were performed for each condition.
(TIF)

**S6 Fig. Zygotes were treated with 60uM MMC or DMSO control and subjected to Caspase-3 activity colorimetric assay.** Optical density was measured at 400nm and is shown in arbitrary units. (A) Lysates were obtained from DMSO and their MMC treated counterparts at 24 hours, and (B) at 48 hours after treatment. The graphs represent the average of 3 biological replicates. Two-sided t test was performed. Error bars represent standard deviation. Individual data points are shown as a jitter. $^{*}$ $p < 0.05$, $^{**}p < 0.01$, $^{***}p < 0.001$.
(TIF)

**S7 Fig. Phylogenetic trees derived using bayesian inference and maximum likelihood based on different alignments.** (A) Topology generated through Bayesian analysis of the full-length sequence alignment for the caspase protein by MAFFT. (B) Topology from maximum likelihood analysis using a MAFFT alignment of the caspase domain (Casc). (C) Topology obtained by maximum likelihood using a muscle alignment of the full-length caspase sequence. (D) Topology obtained by maximum likelihood using a muscle alignment of the caspase domain (Casc).
(TIF)

**S8 Fig. Summary of presence and absence of Caspases in Metazoa. Executioner caspases are the shared and ancestral caspases common to all Metazoa.** Initiator caspases appeared in the common ancestor of cnidarians and bilaterians but were subsequently lost independently in some groups (Nematoda, Hemichordata, Echinodermata and Cephalochordata). Caspase 9 is a specific acquisition of vertebrates.
(TIF)

**S9 Fig. Topologies derived from maximum likelihood analyses of two protein families involved in the apoptosis pathway.** (A) Topology of the Bcl-2 family phylogeny determined by maximum likelihood at the metazoan scale. (B) Phylogeny of Apaf-1 across the metazoan scale, constructed using full-length sequence alignment.
(TIF)

**S1 Text. Detailed information on each of the studies used in cross-species comparative transcriptomic analysis.**
(DOCX)

**S1 Data. FASTA file—Amino-acid alignment used in the phylogenetic analysis.**
(FASTA)

**S1 Table. Genes identified as differentially expressed by both the edgeR and DESeq2 methods.**
(XLSX)

**S2 Table. A full list of all protein sequences used in the BLAST search and phylogenetic analysis for caspase, Apaf-1 and Bcl-2 protein families.**
(DOCX)

**S3 Table. Primer sequences used for RT-qPCR.**
(XLSX)

**S4 Table. Raw data used in S1 Fig.**
(XLSX)

**S5 Table. Raw data used in S3 Fig.**
(CSV)

**S6 Table. Raw data used in S2, S5 and S6 Figs.**
(CSV)

**S7 Table. Raw data used in Figs 3 and 4.**
(XLSX)

## Acknowledgments

The authors thank Dr. Reuven Aharoni (The Hebrew University of Jerusalem) for his technical help.

## Author Contributions

**Conceptualization:** Itamar Kozlovski, Yehu Moran.

**Formal analysis:** Itamar Kozlovski, Adrian Jaimes-Becerra, Magda Lewandowska.

**Funding acquisition:** Yehu Moran.

**Investigation:** Itamar Kozlovski, Adrian Jaimes-Becerra, Ton Sharoni, Magda Lewandowska, Ola Karmi.

**Methodology:** Ola Karmi.

**Supervision:** Yehu Moran.

**Visualization:** Itamar Kozlovski.

**Writing – original draft:** Itamar Kozlovski, Adrian Jaimes-Becerra, Yehu Moran.

**Writing – review & editing:** Itamar Kozlovski, Adrian Jaimes-Becerra, Ton Sharoni, Magda Lewandowska, Ola Karmi, Yehu Moran.

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
