## [Decision Letter · Decision Letter 0]

6 Mar 2024

Dear Dr. Moran,

Thank you very much for submitting your manuscript "Induction of apoptosis by double-stranded RNA was present in the last common ancestor of cnidarian and bilaterian animals" for consideration at PLOS Pathogens. As with all papers reviewed by the journal, your manuscript was reviewed by members of the editorial board and by several independent reviewers. In light of the reviews (below this email), we would like to invite the resubmission of a significantly-revised version that takes into account the reviewers' comments.

Collectively the reviewers identified two issues that should be addressed prior to acceptance for publication. They are: 1.The authors need to make a more convincing case that the observed death is apoptosis and not some other cell death mechanism. 2. More detail should be included regarding the phylogenetic analysis including discussion about the support values for some of the Cas9 clades.

We cannot make any decision about publication until we have seen the revised manuscript and your response to the reviewers' comments. Your revised manuscript is also likely to be sent to reviewers for further evaluation.

Sincerely,

Michael J Layden, Ph.D.

Guest Editor

PLOS Pathogens

Ronald Swanstrom

Section Editor

PLOS Pathogens

Michael Malim

Editor-in-Chief

PLOS Pathogens

orcid.org/0000-0002-7699-2064

Collectively the reviewers identified two issues that should be addressed prior to acceptance for publication. They are: 1.The authors need to make a more convincing case that the observed death is apoptosis and not some other cell death mechanism. 2. More detail should be included regarding the phylogenetic analysis including discussion about the support values for some of the cascade clades.

Reviewer's Responses to Questions

**Part I - Summary**

Reviewer #1: In the manuscript by Kozlovski & Jaimes-Becerra et al the authors describe the establishment of an assay to probe apoptosis in the sea anemone Nematostella vectensis. The authors show that a viral RNA mimic is sufficient to induce apoptosis in Nematostella. They also find an increase in the transcription of known apoptosis-related genes in the organism. Furthermore, they conduct a phylogenetic analysis that shows the conservation of four caspases crucial for apoptosis in animals and an additional cnidarian-specific caspase that is highly upregulated upon the viral RNA mimic exposure. The manuscript combines previously performed transcriptomics analysis (including their own) to give a broader context to apoptosis across the metazoan lineage. The results shown here are overall of great interest as it relates to the emerging notion that innate immune components have a much more ancient evolutionary history than previously thought. Nonetheless, several of the experiments performed raise questions regarding the strength and nature of the signal obtained.

Reviewer #2: In this work, the authors pay interest to the induction of apoptosis by double-stranded RNA in Eumetazoa and in particular in Cnidaria, the sister group of Bilateria. Focusing on the Nematostella vectensis, a cnidarian model organism, the authors show that the injection of a synthetic double stranded RNA mimicking viral infection triggers apoptosis in N. vectensis zygotes as well as in N. vectensis animals. The authors reanalyze RNA-seq data from the literature to show that apoptosis and apoptosis-regulated gene expression is modulated after poly(I:C) injection in N. vectensis and perform a phylogenetic analysis of the diverse caspase genes in Eumetazoa. Finally, the authors re-analyze gene expression datasets from the literature gathered in several model organisms spanning the full Eumetazoa clade to show that apoptosis-related genes are enriched in up/downregulated genes upon poly(I:C) injection.

One strength of the work is to go beyond classical organisms like vertebrates in order to extend what is known about immunity by combining both genomics and experiments. We overall really appreciated to read this manuscript and thank the authors for providing some additional data to the manuscript (e.g the Multiple Sequence Alignment). We think that maybe the phylogenetic analysis could be strengthened and we provide leads to do so.

Reviewer #3: The current article addresses an important topic not only in the field of evolutionary biology but also for immunology studies. The authors stated that while apoptosis as mechanism of defense and immunity has been well study in vertebrates and in some invertebrates, the current state of the art does not allow having a deeper comprehension of this topic at an evolutionary level.

In this study, the author aim at digging deeper into the evolutionary mechanistic aspect of apoptosis by studying this specific biological process in the sea anemone model, Nematostella vectensis that possess all the advantages to tackle such gap in our knowledge. The present study has its roots in a recently published manuscript (Lewandowska et al., 2021) where the authors (same as the present study) showed that Nematostella vectensis possess in it genome two RLR paralogs (named RLRa and RLRb) of the mammalian MDA5 that senses dsRNA, essential actors and sensor mechanism for apoptosis in vertebrate. This published study also showed that RLRb binds to long dsRNA to initiate a functionally conserved innate immune response.

Thus, to go further in the mechanistic aspect of apoptosis in N. vectensis, the authors propose in the current study to determine whether and how dsRNA affect apoptosis in this sea anemone. They show that the dsRNA viral poly(I:C) is sufficient to strongly induce apoptosis in N. vectensis, via a conserved set of genes involved in this process, pointing to a functional conservation of dsRNA-induced apoptosis that existed before the cnidarian-bilaterian split.

While this is an original study with high importance to researchers working in the field of apoptosis, immune response and evolutionary biology and will certainly be of broad interest to the community studying pathogens and pathogen-host interactions, the current state of the paper needs a certain amount of improvements before publication. While the methodology appears appropriate and accurate, some ideas and/or conclusions are not fully supported by the presented data. Thus, the manuscript needs to be completed with some adjustments and additions.

**Part II – Major Issues: Key Experiments Required for Acceptance**

Reviewer #1: 1. The experimental design for some of the experiments is confusing (Figure 1,2). Because of potential variance between experiments, I would expect a consistent positive control with apoptosis inducers across the experiments to account for the wide variance in the arbitrary units of fluorescence. Can it be that the Mock control already induces apoptosis? Please explain the design. A positive control is also particularly relevant for the detection in the cas-3 assay which in some time points exhibit a high variance compounded with only 3 replicates.

2. Regarding the timeline for the apoptotic response to treatment, it would be helpful to receive context from previous experiments (other studies) as to the typical temporal response after inoculation of Poly(I:C).

3. Why was the negative control for apotracker performed using a mock transfection but the colorimetric assay for cas-3 was performed with NaCl, or am I missing something? Please clarify.

4. Regarding the phylogenetic analysis, in order to put the caspases into a larger phylogenetic context, I suggest to add annotations of well-studied representatives of other phylogenetic clades. This would greatly help follow the evolutionary signal (the annotation is currently a bit cumbersome).

5. It is not totally convincing that the authors actually observe apoptosis and not another kind of immune-related cell death. The authors discuss this point shortly in the discussion, yet I find this not sufficient. The authors may consider analyzing their transcriptomics data and genomics data for pyroptosis (or necroptosis) related genes. At least they should further discuss this possibility.

6. Relating to the previous point, the manuscript could be strengthened if the actual morphological phenotype was observed (by microscopy), diminishing the possibility of other mechanisms of cell death being at play which is followed by the apotracker expeirments.

7. I believe that in terms of organization, the manuscript might have been more accessible and seamless if the results would begin with the transcriptomics analysis of the previous paper

Reviewer #2: • We would ask the authors to provide a bit more information regarding the phylogenetic analysis:

o In the results, please describe in one sentence the procedure of identifying Caspase homologs in Cnidarians (prior to diving into the phylogenetic analysis). In particular, in which clade did the authors search for Caspase homologs ? How many hits were obtained in total ? This information could be provided in one or two sentences. Then, in the methods, please provide more information about the database in which you searched for homologs of human caspases : how many genomes ? how many hits on average per genome within each taxon ?

o Please add a supplementary figure being a phylogenetic tree of all animals (or all Eumetazoa) showing in which taxa each Caspase is present (e.g. using one outer ring/colour per known Caspase ). This would really help the reader to understand which Caspase is present in which taxon (e.g. to understand very easily that Casp9 appeared only later in animal evolution.

o How did the authors define Clades I to VI in the phylogenetic tree ? According to UFBoostrap values, some clades are not very well supported (when UFBoostrap < 80, the authors should consider that a clade is not well supported at all whereas only clades with UFBoostrap > 95 should be considered as well supported).

o Was the Multiple Sequence Alignment built on the full length of each protein ? We would ask the authors to also compute a Multiple Sequence Alignment only on the Caspase (CASc) domain. This would first allow to get a better MSA (when looking at the one the authors provided, it seems that best aligned region is really the one corresponding to this domain vs. the N-Terminal part of proteins). With a better alignment, one also generally gets better support values for the different clades which would make the clade I-IV definition more convincing. Finally, it is always interesting to compare trees obtained at the protein level vs. at the domain level.

o Optionally, the authors could also use alternative aligners such as Muscle (with the super5 algorithm) in order to test if the tree topology is robust to this type of change. The authors could provide in supplementary figure the trees obtained using this aligner in order to show the robustness of their results.

Reviewer #3: No Major Issues are present in the current study.

**Part III – Minor Issues: Editorial and Data Presentation Modifications**

Reviewer #1: - Lines 243 – 252 belongs in the discussion

- Organization of figure 2: for comparison a,b,c should be in one line

Reviewer #2: • Figure 1

o Figure 1 could be smaller (e.g. panels B and D could be reduced)

o What are the three pairs of curves representing in Figures 1A and 1C ? We suppose this is three time points (6h, 24h and 48h) but the information is not provided in the Figure description. The authors should provide this information.

o The same y-axis should not be used for the three pairs of curves in Figures 1A and 1C and ticks should be added (even though this is only 0 and 1 for normalized

• Page 7 “whereas caspase activity […] did not reach a statistical significance at 48hpi” : what do the authors conclude from that ? Do they expect that the response should decrease after 48h ?

• Page 7 “which is not orthologous of TLR3” : Missing reference.

• Paragraph “dsRNA induces apoptosis in N. vectensis injected animals” : it feels like this paragraph lacks an introduction and a conclusion to really link with what comes before and after. What does the paragraph brings to the whole ? I think stating it clearly will help the reader to follow the story.

• Figure 3 : The analyzed dataset comes from Lewandowska et al., right ? Please specify it in the Figure legend.

• Figure 3A : It is unclear to us why some genes appear several times in the volcano plot (e.g. Caspase 3/6/7 or Bcl2-like). Please precise this in the Figure legend.

• In Figure 4, the authors could indicate which prefix corresponds to which species with some taxonomic information (e.g. edi = Exaiptasia diaphana) in order to guide the reader.

• Page 12 : “dsRNA induces a conserved antiviral response pathway across metazoans” : isn’t it Eumetazoans instead ? Metazoans is more broad (it includes e.g. Porifera).

Reviewer #3: General comments:

The references of all products used in this study need to be added and homogenized in the entire manuscript. All figure names need to be corrected in regard to the name of the figure in the figure legends and homogenized in the entire manuscript.

P3 Introduction

L102 - Nevertheless – Typo

L111 – Syntax:

Replace: “In this study” by “In the current study” as the first form seems to refer to the previous study performed by the authors (ref 52).

L113-114: “and uncovered a conserved network of genes involved in this process”,

Com

---

## [Decision Letter · Decision Letter 1]

6 Jun 2024

Dear Dr. Moran,

We are pleased to inform you that your manuscript 'Induction of apoptosis by double-stranded RNA was present in the last common ancestor of cnidarian and bilaterian animals' has been provisionally accepted for publication in PLOS Pathogens.

Thank you for dutifully responding the reviewers requests. One minor comment was suggested that protein accession rather than gene accession numbers are used in the supplementary file. If it is not too much trouble, I would consider changing that in the final version of the supplementary material.

Best regards,

Michael J Layden, Ph.D.

Guest Editor

PLOS Pathogens

Ronald Swanstrom

Section Editor

PLOS Pathogens

Michael Malim

Editor-in-Chief

PLOS Pathogens

orcid.org/0000-0002-7699-2064

Thank you for dutifully responding the reviewers requests. One minor comment was suggested that protein accession rather than gene accession numbers are used in the supplementary file. If it is not too much trouble, I would consider changing that in the final version of the supplementary material.

Reviewer Comments (if any, and for reference):

Reviewer's Responses to Questions

**Part I - Summary**

Reviewer #2: We thank the authors for the constructive review process. They adressed our comments.

One minor point in Supp File S2, would be good to add protein accessions and not genomes accessions.

Congratulations to the authors.

**Part II – Major Issues: Key Experiments Required for Acceptance**

Reviewer #2: (No Response)

**Part III – Minor Issues: Editorial and Data Presentation Modifications**

Reviewer #2: (No Response)

PLOS authors have the option to publish the peer review history of their article (what does this mean?). If published, this will include your full peer review and any attached files.

Reviewer #2: **Yes: **Hugo Vaysset and Aude Bernheim

---

## [Editor Report · Acceptance letter]

29 Jun 2024

Dear Dr. Moran,

We are delighted to inform you that your manuscript, "Induction of apoptosis by double-stranded RNA was present in the last common ancestor of cnidarian and bilaterian animals," has been formally accepted for publication in PLOS Pathogens.

Best regards,

Michael Malim

Editor-in-Chief

PLOS Pathogens

orcid.org/0000-0002-7699-2064